# Cysteamine/Cystamine Exert Anti-*Mycobacterium abscessus* Activity Alone or in Combination with Amikacin

**DOI:** 10.3390/ijms24021203

**Published:** 2023-01-07

**Authors:** Ivana Palucci, Alessandro Salustri, Flavio De Maio, Maria del Carmen Pereyra Boza, Francesco Paglione, Michela Sali, Luca Occhigrossi, Manuela D’Eletto, Federica Rossin, Delia Goletti, Maurizio Sanguinetti, Mauro Piacentini, Giovanni Delogu

**Affiliations:** 1Department of Laboratory Sciences and Infectious Diseases, Fondazione Policlinico Universitario “A. Gemelli”, 00168 Rome, Italy; 2Dipartimento di Scienze Biotecnologiche di Base, Cliniche Intensivologiche e Perioperatorie—Sezione di Microbiologia, Università Cattolica del Sacro Cuore, 00168 Rome, Italy; 3Dipartimento di Biologia, Università degli Studi di Roma “Tor Vergata”, 00173 Rome, Italy; 4Department of Epidemiology and Preclinical Research, National Institute for Infectious Diseases Translational Research Unit, IRCCS ‘Lazzaro Spallanzani’, 00149 Rome, Italy; 5Mater Olbia Hospital, 07026 Olbia, Italy

**Keywords:** *Mycobacterium abscessus*, transglutaminase-2, cystamine, cysteamine, host-directed therapies

## Abstract

Host-directed therapies are emerging as a promising tool in the curing of difficult-to-treat infections, such as those caused by drug-resistant bacteria. In this study, we aim to test the potential activity of the FDA- and EMA-approved drugs cysteamine and cystamine against *Mycobacterium abscessus*. In human macrophages (differentiated THP-1 cells), these drugs restricted *M. abscessus* growth similar to that achieved by amikacin. Here, we use the human ex vivo granuloma-like structures (GLS) model of infection with the *M. abscessus* rough (MAB-R) and smooth (MAB-S) variants to study the activity of new therapies against *M. abscessus*. We demonstrate that cysteamine and cystamine show a decrease in the number of total GLSs per well in the MAB-S and MAB-R infected human peripheral blood mononuclear cells (PBMCs). Furthermore, combined administration of cysteamine or cystamine with amikacin resulted in enhanced activity against the two *M. abscessus* morpho variants compared to treatment with amikacin only. Treatment with cysteamine and cystamine was more effective in reducing GLS size and bacterial load during MAB-S infection compared with MAB-R infection. Moreover, treatment with these two drugs drastically quenched the exuberant proinflammatory response triggered by the MAB-R variant. These findings showing the activity of cysteamine and cystamine against the R and S *M. abscessus* morphotypes support the use of these drugs as novel host-directed therapies against *M. abscessus* infections.

## 1. Introduction

Non-Tuberculous Mycobacteria (NTM) groups mycobacterial species other than *Mycobacterium tuberculosis* (*M.tb*) complex and *M. leprae* and includes more than 150 officially recognized species, commonly found in soil and water [1]. These ubiquitous species may also be found in household water systems, potting soil, and medical equipment [2]. NTM includes species that can cause diseases in humans and, depending on several features, primarily the host-immune status, they can cause a spectrum of conditions that in several cases can be life-threatening. Among the NTM species, *Mycobacterium avium* complex (MAC), *M. intracellulare* complex (*M. intracellulare* and *M. chimaera*), and *Mycobacterium abscessus* are the most common cause of disease, accounting for more than 90% of the total cases reported [3].

Despite significant NTM interspecies variability in overall virulence potential, pathogenetic mechanisms, and intrinsic drug resistance, therapeutic regimens against NTM are often empirical due to the lack of solid guidelines [4]. Standardized regimens are developed only for the most common NTM as MAC, *M. intracellulare*, *M. xenopi*, *M. kansasii*, and *M. abscessus* and combine several drugs that are administered for months [4,5]. *M. abscessus* is an emerging pathogen that causes lung diseases primarily in cystic fibrosis (CF) patients [6], it is overall insensitive to “standard” antituberculosis drugs defined by the World Health Organization (WHO) and susceptibility is limited to few antibiotics as amikacin (AMK) and macrolides. Indeed, the combined use of clarithromycin and amikacin forms the cornerstone of the recommended regimens against *M. abscessus* infections, which usually include other antibiotics such as imipenem, cefoxitin, or tigecycline [7,8].

Despite the implementation of these aggressive regimes, outcomes are overall unsatisfactory with cure rates below 50% and the impossibility to eradicate *M. abscessus* infection [5,9]. Hence, there is an urgent need to develop more effective and safe regimens against *M. abscessus*. The strategy to repurpose or reformulate existing antibiotics known to be active against *M.tb* may be problematic and perhaps futile due to the too often neglected genetic and physiological differences between *M. abscessus* and the tubercle bacillus [10].

Host-directed therapies (HDTs) are emerging as an attractive strategy in the fight against infectious agents and aim either at supporting the host’s antimicrobial activities or at dampening the inflammatory and tissue-damaging responses associated with infections [11,12,13]. HDTs usually rely on the use of drugs that have been already approved to treat chronic diseases (such as diabetes, cancer, and hypertension), manipulate the metabolism and/or immune cell function to optimize the pro-inflammatory response or modify the tissue physiology, and are used as an adjunct therapy to antibiotic regimens [14].

We have recently shown that genetic or pharmacologic inactivation of transglutaminase 2 (TG2), a pleiotropic enzyme involved in many key cellular processes (proteostasis, autophagy, death/survival) [15] enhances macrophage anti-mycobacterial properties by interfering with the LC3/autophagy homeostasis and associated downstream processes, including regulation of the Type I Interferon response [16,17,18]. Indeed, treatment of *M.tb*-infected macrophages or peripheral blood mononuclear cells (PBMCs) with two TG2 inhibitors, cysteamine or cystamine, significantly impaired *M.tb* intracellular replication [16,17].

Cystamine, a symmetric organodisulfide, is readily reduced to cysteamine within the body [19]. Cysteamine has been approved by the Food and Drug Administration (FDA) and the European Medicines Agency (EMA) for the treatment of nephropathic cystinosis [20]. Interestingly, treatment with cysteamine re-establishes bacteria clearance in an experimental model of CF [21], and both cysteamine and cystamine showed immune-modulatory effects against SARS-CoV-2 infection [22,23].

Based on these findings, we hypothesize that these two drugs may be effective against NTM infections. In this study, we tested this hypothesis by assessing the activity of cysteamine and cystamine against the emerging pathogen *M. abscessus* using the ex vivo model of Granuloma-like structures (GLS) that allows a more complete assessment of HDTs [17,24].

## 2. Results

Treatment with cysteamine or cystamine enhances the anti-*M. abscessus* activity of human macrophages. As a follow-up to our previous study in *M.tb* [16], we aimed to investigate the potential usefulness of TG2 as a target for anti-*M. abscessus* regimens. For this purpose, human macrophages (differentiated THP-1 cells), were infected with a clinical strain of *M. abscessus*, and four hours post-infection cystamine was added to the infected macrophages; forty-eight hours later, the bacterial burden was assessed by CFU counting (Figure 1A). Treatment with cystamine restricted intracellular replication in human macrophages compared to the untreated control, similar to what was previously observed for *M.tb* [17]. This result provides evidence supporting cystamine as a potential target for HDTs against *M. abscessus*.

Like *M. avium* or *M. smegmatis*, *M. abscessus* displays two distinct morphotypes on solid media [25]: the smooth (S) variant, non-cording but motile and biofilm-forming; and the rough (R) variant, cording but non-motile and non-biofilm-forming [26,27]. We isolated the S and R variants by subculturing the clinical *M. abscessus* strain and first assessed the activity of cysteamine or cystamine to already grown cultures. As we and others have observed with *M.tb* [17,28], treatment with the two drugs of the two *M. abscessus* variants did not affect their bacterial viability, nor did administrations of these two drugs improve the activity of amikacin, indicating no direct effects of these two TG2 inhibitors on *M. abscessus* (Figure 1B,C). Moreover, as shown in Table 1, the addition of cysteamine or cystamine to commonly used antibiotics did not affect the MIC of the two *M. abscessus* variants, further supporting the lack of any direct activity of these two drugs against mycobacteria (Appendix A), at least at the concentrations used in these settings (400 µM cystamine, 800 µM cysteamine).

To further investigate the potential usefulness of cysteamine and cystamine, we infected human macrophages (differentiated THP-1 cells), with the two different *M. abscessus* variants (MAB-S and MAB-R), using different MOIs as suggested previously by other authors [6], given their different virulence potential. Then infected cells were treated with cysteamine and cystamine at concentrations compatible with those commonly achieved in vivo [16] (400 µM cystamine, 800 µM cysteamine) [21,29], alone, or in combination with a commonly used antibiotic, amikacin, administered at the MIC (4 μg/mL) to measure the potential of the HDTs to support antibiotic treatment. As shown in Figure 2A,B, cysteamine and cystamine were able to restrict both *M. abscessus* variants in macrophages similarly to what was achieved by amikacin administered at the MIC. Interestingly, combined use of cysteamine and cystamine with amikacin, resulted in an enhanced restriction of *M. abscessus* in macrophages, providing a reduction of 1.01 ± 0.21 for MAB-S and 0.80 ± 0.06 for MAB-R Log CFU/10^6^ cells compared to amikacin alone. Taken together, these results indicate that cysteamine and, most remarkably, cystamine, can induce anti-mycobacterial activity in infected macrophages against *M. abscessus* and their use in combination with a commonly used antibiotic can warrant a significantly enhanced effect.

Treatment with cysteamine and cystamine improves containment of the *M. abscessus* infection in the GLS assay. The ex vivo model of GLSs is emerging as a valuable tool to investigate the pathogenesis of mycobacterial infections and assess the activity of new drugs, most importantly those targeting the host [9,17,30]. PBMCs, obtained from *Mtb*-uninfected healthy donors, were infected with the MAB-R and -S variants, obtained from the *M. abscessus* clinical isolate. At day 3 p.i., the drugs were added and GLSs were monitored over time; at day 12 p.i., the number and size of GLSs were evaluated by microscopic analysis and bacterial replication measured by CFU counting. As shown in Figure 3, treatment with amikacin, cysteamine, or cystamine resulted in a reduction in the number of total GLSs per well compared to the untreated group. Interestingly, the average area of the GLS surface was significantly reduced when infected PBMCs were treated with a combination of amikacin and cysteamine.

As shown in Figure 4A,B, treatment of PBMCs infected with the MAB-S variant with cysteamine or cystamine achieved a reduction in CFUs that was more robust than that achieved by treatment with amikacin. Interestingly, combined treatment of amikacin with cysteamine or cystamine resulted in an enhanced antimicrobial effect, reaching a ≈70% reduction in CFUs (Figure 4B). We observed no differences between the S and R variants in terms of cytotoxicity calculated with LDH assay (Appendix A). Treatment with amikacin of PBMCs with the MAB-R variant warranted a robust reduction in CFUs (≈70%), and although treatment with cysteamine and cystamine led to a similarly robust reduction, the combined use of the two drugs with amikacin did not result in any enhanced anti-mycobacterial activity (Figure 4C,D). Taken together, the results obtained in the GLS model indicate that cysteamine and cystamine can exert anti-mycobacterial activity against *M. abscessus*, with differences between the S and R variants.

Infection of PBMCs with the MAB-R variant triggers a powerful inflammatory response that is quenched following treatment with cysteamine and cystamine. Infecting PBMCs with *M. abscessus* to generate the GLS model offers the opportunity to have multiple types of cells that, in part, recapitulate the complex interplay between mycobacteria and peripheral host cells [31]. In addition to monitoring for the number of GLS per well and their size, the experimental model allows the assessment of the secretion of chemokines and cytokines in the supernatant, providing relevant information on the capacity of different treatments to modulate host immune responses. As shown in Figure 5, we measured the concentrations of selected cytokines and chemokines released in the supernatant following infection and treatment in the GLS model. Infection with the MAB-R variant triggered higher levels of IL-1β (32,132 ± 0.80 pg/mL), IL-6 (14,862 ± 0.61 pg/mL), TNF-α (49,896 ± 1.89 pg/mL) and, though to a lesser extent, IL-8 (115.314 ± 2.68 pg/mL) compared with infection with the MAB-S variant (IL-1β 57. 5 pg/mL ± 0.2, IL-6 188 pg/mL ± 0.6; TNF- α 1658 pg/mL ± 0.93, IL-8 63,680 pg/mL ± 1), in line with previous findings [32]. Treatment with amikacin, cysteamine, and cystamine, or their combination with amikacin did not dramatically affect the concentration of IL-6, TNF-α, and IL-8 in PBMCs infected with the MAB-S variant. Conversely, a dramatic reduction in the concentrations of the pro-inflammatory cytokines was observed when the MAB-R variant was treated with amikacin or the TG2 inhibitors (Figure 5A–D), with the only exception being the enhanced secretion of IL-8 (Figure 5E,F) in PBMCs infected with the MAB-R strain and treated with cystamine. Levels of IFN-I were overall low following infections of PBMCs with the MAB-R and -S variants, and treatment with amikacin, cysteamine, and cystamine or their combination did not significantly modulate the concentration of type I IFN (Appendix A). Conversely, infection with MAB-R triggered secretion of IL-1β (≈30.000 pg/mL) that was dramatically reduced following treatment with amikacin (≈30 pg/mL). Treatment with cysteamine or cystamine of PBMCs infected with MAB-R led to levels of IL-1β slightly higher than those observed in the amikacin-treated group. Interestingly, infection of PBMCs with MAB-S did not lead to high levels of IL-1β, although treatment of these infected cells with cysteamine and cystamine warranted a higher level of IL-1β in the supernatant compared to the untreated and amikacin-treated groups.

## 3. Discussion

HDTs are emerging as a promising tool in the curing of difficult-to-treat infections, as in the case of infections caused by drug-resistant bacteria [33]. Therapy of NTM infections can be challenging due to the intrinsic resistance of these bacteria to most antibiotics, the need to use multiple drugs simultaneously, and the rapid emergence of adaptive drug resistance [34]. In this study, we provide experimental evidence in relevant in vitro models that cysteamine and cystamine, two molecules that inhibit TG2 and have an impact on many intracellular and extracellular processes, can be effective in enhancing host antimicrobial responses and exert enhanced activity against *M. abscessus* when administered in combination with aminoglycosides.

*M. abscessus* is a rapidly growing mycobacteria that can cause soft tissue infections and pulmonary infections primarily in CF patients and patients with underlying chronic lung conditions such as bronchiectasis [31,35]. Therapeutic regimens against *M. abscessus* are poorly effective due to intrinsic and acquired mechanisms of resistance [36]. *M. abscessus* can switch from the glycopeptidolipids (GPL)-rich S variant, which is the *M. abscessus* infectious form, to the GPL-deficient R variant [37], which is more virulent due to the ability to promote bacterial extracellular replication that results in the forming of cording-like structures that cannot be phagocytosed by macrophages and can promote extensive inflammation and tissue damage [31]. The transition from the S to the R variant is an irreversible process that occurs during infection [38], making the more-virulent R variant dominant. Hence, it is important to deploy therapeutic regimens that are active against the S variant, which can persist in vivo for a long time and corresponds to the bacterium transmissible unit, and the R variant, which is responsible for the exacerbation of the disease. Cysteamine and cystamine showed activity, alone or in combination with an aminoglycoside, against the S and R forms. It can be speculated that the activity of cysteamine and cystamine, similar to what we observed with *M.tb*, may result from the ability of these TG2-inhibitors to modulate the autophagic process which plays an important role during the infections of both variants [1]. However, cysteamine and cystamine can interfere with many intracellular and extracellular processes and we cannot exclude other mechanisms involved in their anti-mycobacterial activity, as highlighted by the results obtained in the PBMCs model of infection, where the intracellular signaling pathways are affected by the chemokine and cytokine milieu [18,22]. Specific characterization of the activity of the two HDTs on different cell types goes beyond the scope of this study and we hope that the results provided can stimulate further investigations on this topic. Of note, our experiments do not lend support to the direct activity of cysteamine against mycobacteria, whereas previous studies showed the ability of cysteamine to inhibit *M. abscessus* and other bacterial species [39,40], though at concentrations higher than that used in our experimental settings.

The GLS model is a reliable and valid tool to assess the activity of any anti-mycobacterial drug and primarily those used in HDTs since it considers the complex interplay between mycobacteria and different types of peripheral host immune cells [17]. Infection of PBMCs with *M. abscessus* can be maintained for up to 12 days in vitro, offering the possibility to measure bacterial persistence and replication over a wide time range and the host response by measuring relevant chemokines and cytokines. The MAB-R variant was able to elicit much higher levels of IL-6, TNF-α, and IL-1β compared to the S variant, in line with the superior inflammatory potential of the former [32]. Indeed, the lack of GPL on the surface of the MAB-R variant exposes potent pro-inflammatory stimulants such as phosphatidyl-myo-inositol mannosides and lipoproteins which are masked in the S variant [31,32]. The enhanced inflammatory potential of the R variant accounts for the extensive tissue damage and superior virulence observed in in vitro and in vivo models and for the exacerbated clinical conditions observed in patients where infection is dominated by the R variant [31,41]. Treatment with amikacin of PBMCs infected with the MAB-R led to a dramatic reduction in the secretion of IL-6, TNF-α, and IL-1β inflammatory responses, suggesting that bacterial burden is a key event in triggering the secretion of inflammatory cytokines. Likewise, a dramatic reduction in the release of inflammatory cytokines was observed following treatment with cysteamine and cystamine. However, PBMCs infected with the MAB-R variant and treated with cystamine/amikacin showed higher levels of IL-8 compared to similarly infected PBMCs and treated with the cystamine or amikacin alone, or with the combination of amikacin and cysteamine. IL-8 is secreted by macrophages and epithelial cells in response to TLR activation and promotes chemotaxis, primarily of neutrophils. Importantly, as described in the zebrafish model by Bernut A et al. [42], IL-8 ablation correlated with reduced larval survival, with increased S and R loads and with the numerous large abscesses, suggesting that the absence of neutrophils at the site of infection may be deleterious for the host. We plan to investigate the consequences in terms of cellular infiltration and tissue damage, in addition to bacterial tissue burden, in animals infected with *M. abscessus* and treated with cystamine and amikacin.

The S to R switch alters the outer layer of the *M. abscessus* cell envelope, with loss of GPL and increased expression of lipoproteins. These lipoproteins act as strong TLR-2 ligands that modulate host innate immune responses [31]. Indeed, the production of TNF-α in response to mycobacterial infection elicits the early secretion of chemokines that orchestrate the recruitment of leukocytes to form granulomas [43]. Although the pathogenetic mechanism of the MAB-R variant, compared to the S variant, resembles that of *M.tb*, the R variant utilizes a different escape mechanism, with the bacterial burden overloading phagosome capacity, leading to disruption of membranes and causing cell necrosis, followed by IFN-I production and cell-to-cell spread [44]. The increased IFN-I induction observed in response to infection [45] with the MAB-R variant was associated with a corresponding increase in TNF-α and intracellular death induced by enhanced macrophage nitric oxide production. In contrast, deficiencies in TNF-α release and the lack of IL-8 neutrophil chemoattraction result in anomalous granuloma formation [46]. A recent study showed that Type I IFN signaling induced by the MAB-R variant contributes to virulence via cell-to-cell spread in Type I IFN-dependent mechanism through the cGAS-STING-IRF3 pathway [44], suggesting that Type I IFN contributes to the enhanced virulence of the MAB-R variant by providing a niche for their survival within the macrophage.

The MAB-R variant triggered higher levels of IL-1β compared to the S variant, and treatment with cysteamine and cystamine drastically reduced IL-1β in the PBMC model of infection. ROS production in macrophages may be important for inflammasome activation and IL-1β production via NLRP3 activation, where caspase-1 cleaves pro-IL-1 into active IL-1β [47], contributes to prime and activate adaptive immunity and orchestrate the inflammatory responses during mycobacteria infection [48]. As already suggested, IL-1β is a potential target of HDT to improve the immune-mediated pathology associated with inflammation in individuals with TB [48]. Abnormal cytokine networks impair anti-TB immunity and facilitate the formation of caseous granulomas [49]. Secretions of cytokines and ROS from immune cells that defend the host against *M.tb* infection are crucial for the formation of normal and compact granulomas to control infection. In individuals with diabetes, immunodeficient conditions, or undergoing biological treatments, abnormal cytokine networks or ROS production can lead to excessive levels of IL-1β in the tissue and cause the formation of neutrophilic caseous granulomas [50]. Hence, treatments based on the depletion of IL-1β may alleviate destructive tissue damage and facilitate the emergence of an effective immune response during *M. abscessus* infections [51].

Elucidating the immunological mechanisms governing the response against the MAB- R and -S variants is essential for the development of HDTs. Enhancing antimicrobial activity while restoring a balanced immune response against this emerging pathogen is crucial, primarily in vulnerable patients such as CF patients, who are most affected by this pathogen and that can contribute to the spread of the infection [29]. In fact, while infection is established due to the ability of the S variant to minimally induce host immune responses, the transition to the R variant prompts inflammatory responses that promote tissue damage that then leads to overt disease [44]. Our data suggest that cysteamine and cystamine may be effective HDTs by enhancing antimicrobial responses against the S and R variants and by alleviating excessive inflammation promoted by the R variant, potentially leading to more effective and shortened therapeutic regimens.

In conclusion, the limited number of effective antibiotics currently available and the ability of these bacteria to rapidly evolve resistance to any new antibiotic prompts the need to develop new control strategies. Among the options to be considered, the use of cysteamine and cystamine, or other effective HDTs, may be very important since this can reduce bacterial burden even when adaptive resistance emerges, thereby mitigating disease and transmission. The fact that these two HDTs are also active against intracellular MAB-S variant is of relevance since it may reduce transmission efficiency. Future studies in relevant animal models will clarify the potential usefulness of cysteamine and cystamine as a support therapy against *M. abscessus* infections.

## 4. Material and Methods

Bacterial Strains. The strain used in the present study is the *M. abscessus* clinical strain, named FPG2018, isolated from the sputum of a CF patient hospitalized at IRCCS- Fondazione Policlinico A. Gemelli Università Cattolica del Sacro Cuore di Roma. The original *M. abscessus* clinical strain displayed smooth (S) colonies on agar medium 7H11 Middlebrook (Difco, Spark, MD, USA), but after a couple of passages on agar medium, natural rough (R) colony mutants appeared. MAB-S and R variants were identified using polymerase chain reaction and sequencing of the *hsp65* gene [52,53]. All strains were grown in 7H9 (Difco, Spark, MD, USA) supplemented with 10% (vol/vol) oleic acid-albumin-dextrose-catalase (OADC; Difco, Spark, MD), with 0.2% glycerol (Microbiol, Cagliari, Italy) and 0.05% Tween 80 (Sigma-Aldrich, St. Louis, MO, USA) at 37 °C for 4/5 days. Mycobacterial cultures were harvested at the late log phase, glycerol was added at 20% final concentration, and 1 mL aliquots were stored at −80 °C. These stocks were enumerated by CFU counting and infection of differentiated THP-1 cells or PBMCs was carried out with the proper dose of bacteria starting from these enumerated glycerol stocks [54]. MIC determination was carried out by the broth microdilution assay using the SLOWMYCO system Sensititre 96-well plate (Thermo Scientific, Waltham, MA, USA) following instructions provided by the manufacturer. Cysteamine or cystamine was added at the final concentrations of 800 μM and 400 μM, respectively.

Study Participants. The PBMCs were derived from healthy donors. Participants were recruited among people who had recently tested negative for QFT negative, not vaccinated with BCG, male, Caucasian, and aged between 30 and 35 years. Written informed consent was obtained from each donor (ID:3715/2021).

Human cells and culture media. THP-1 cells were grown in complete RPMI 1640 medium in a tissue culture flask with a minimum volume of 20 mL and a maximum volume of 50 mL and were incubated in an atmosphere of 95% air and 5% carbon dioxide (CO_2_) at 37 °C. The cell density was kept between 0.25 million and 1 million cells/mL. For culturing of THP-1 cells, RPMI 1640 medium supplemented with 5% fetal bovine serum (FBS), 2% glutamine, 1% nonessential amino acids, and 1% penicillin-streptomycin was used (complete RPMI medium).

Peripheral blood mononuclear cells (PBMCs) were obtained from healthy donors and were isolated by density gradient centrifugation. HEK-Blue™ IFN-α/β cells (InvivoGen^®^, San Diego, CA, USA) selected antibiotics (Blasticidin, Normocin, and Zeocin) and SEAP substrate (QUANTI-Blue™) were purchased from InvivoGen^®^ (San Diego, CA, USA). HEK-Blue cells were seeded in 96-well plates at a concentration of 50,400 cells/well. Twenty microliters of PBMCs supernatants were added to the corresponding wells and incubated at 37 °C and 5% CO_2_ for 24 h. After the incubation period, 20 μL of supernatant was transferred into a 96-well plate with 180 μL of QUANTI-Blue™ ready to use solution (prepared as indicated by the manufacturer). Finally, O.D. at 630 nm was measured using a spectrophotometer [55].

*M. abscessus* infection assay in human cells. Before infection, THP-1 cells (5 × 10^5^/well) were cultured on plates (Thermo Scientific™ Nunc, USA) with 20 nM PMA (Sigma-Aldrich, St. Louis, MO, USA) for 48 h to induce their differentiation into macrophages, then washed three times with PBS, and maintained in 5% FBS [17]. At first, we evaluated the intracellular growth of the original *M. abscessus* and both variants in human macrophages by infecting cells at an MOI of 1:10 for 2 h. However, we observed important differences in the mycobacterial uptake for the S and R variants (up to a one log10 difference for the R variant) that prompted us to use different MOIs for the two variants [6]. The macrophages were infected with the S variant at MOI (bacteria per macrophage) 1:1, while a MOI 1:10 was used for the R variant [56]. Two hours post-infection (p.i.), cells were washed twice with sterile phosphate-buffered saline (PBS) to remove extracellular bacteria, lysed in 0.01% Triton-X100 (Sigma-Aldrich, St. Louis, MO, USA) and intracellular bacterial loads determined by CFU counting [17]. Treatment of infected cells was started at 4 h p.i. by adding cysteamine (800 µM), cystamine (400 µM), or amikacin (4 μg/mL, Sigma-Aldrich, St. Louis, MO, USA) alone or in combination as indicated [16,17].

Activity of the drugs against MAB- S and R variants in axenic culture. The direct activity of the two TG2 inhibitors was evaluated on the two *M. abscessus* morphotypes. Tubes (15 mL) of 7H9 broth (Difco, Spark, MD, USA) containing ADC and the different drugs alone or in combination were inoculated with 1 × 10^4^ CFU/mL of mycobacteria and incubated for 2 days at 37 °C with shaking (100 rpm). Bacteria were enumerated at 0, 6, 12, 18, and 48 h by plating serial dilutions on Middlebrook 7H11 agar supplemented with 0.05% Tween 80 and 10% OADC (oleic acid, dextrose, catalase, and bovine albumin) plates. CFUs were enumerated after an additional 5–7 days of incubation at 37 °C. All assays were performed in duplicate.

Infection with *M. abscessus* in the Granuloma-Like Structure (GLS) model. PBMCs were obtained from healthy donors and isolated as described above, counted, and immediately infected with both variants, MAB-S at MOI of 1:1 and MAB-R at MOI of 1:10 with respect to monocytes and incubated for up to 3 days to let the GLS begin to form. At this time, treatments with the selected drugs were started and analysis was performed as previously described [17,24]. The analysis of the stage of GLS has been done daily by using an inverted light microscope. At least 12 separate fields per sample were analyzed to measure the area and total number of GLS [24]. Infected GLS were lysed at 12 days post-infection, as described previously [17].

Cytokine ELISA. IL-6, IL-8, TNF-α, and IL-1β supernatants levels were quantified by single-plex ELISA (Multi-cytokine test for ELLA, Bio-Techne, Minneapolis, MN, USA). Moreover, 10 µL of supernatants were used to measure LDH concentrations using the LDH-Cytotoxicity Assay Kit as indicated by the manufacturer (Sigma-Aldrich).

Statistics. Data were analyzed using the GraphPad Prism software, version 7.02 for Windows (GraphPad Software, San Diego, CA, USA). All experiments were performed at least three times in triplicate. The results of the experiments carried out in axenic cultures were evaluated using one-way ANOVA with Dunnett’s multiple-comparisons test against an untreated control. The statistical significance in the growth of the MAB-R and -S variants in axenic cultures and GLSs analysis was evaluated with two-way ANOVA with Dunnett’s multiple-comparisons test against an untreated control. The healthy donors used for GLS formation were adult (18–45 years of age), uninfected, and non-vaccinated. Differences were considered significant if *p*-values were ≤0.05. The differences in intracellular replication of the MAB-R and -S variants were evaluated using one–way ANOVA and Tukey’s multiple comparisons test.

## Figures and Tables

**Figure 1 ijms-24-01203-f001:**
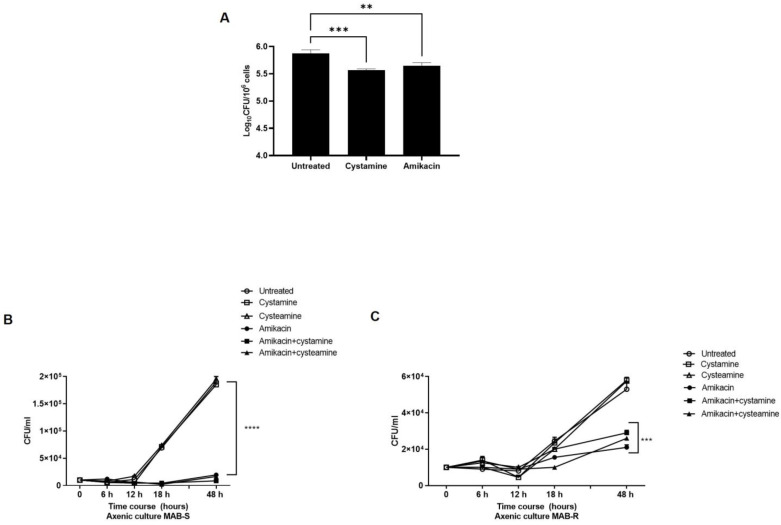
Inhibition of TG2 enhances anti-mycobacterial activity in human macrophages. (**A**) Human macrophages (differentiated THP-1 cells) were infected at an MOI of 1:10 with the *M. abscessus* clinical strain FPG2018 and 4 h post-infection (p.i.) cells were washed and fresh media added without or with cystamine (400 µM) and amikacin (4 µg/mL). Intracellular mycobacteria were determined by CFU counting at 2 days post-infection (** *p* < 0.01 and *** *p* < 0.005, **** *p* < 0.001 compared with the untreated group using one-way ANOVA followed by Dunnett’s multiple comparison test). Panels (**B**,**C**) show the viability of MAB-S and -R variants in axenic culture, treated with amikacin (4 µg/mL), cystamine (400 µM), or cysteamine (800 µM), and the combinations of these with amikacin (using the same concentrations as the individual treatments). Aliquots were taken at indicated times and plated to determine CFUs. The average with SD is plotted (*n* = 3). Values are expressed as a mean of three independent experiments. Data were analyzed by one-way ANOVA with Dunnett’s multiple comparisons test against untreated control.

**Figure 2 ijms-24-01203-f002:**
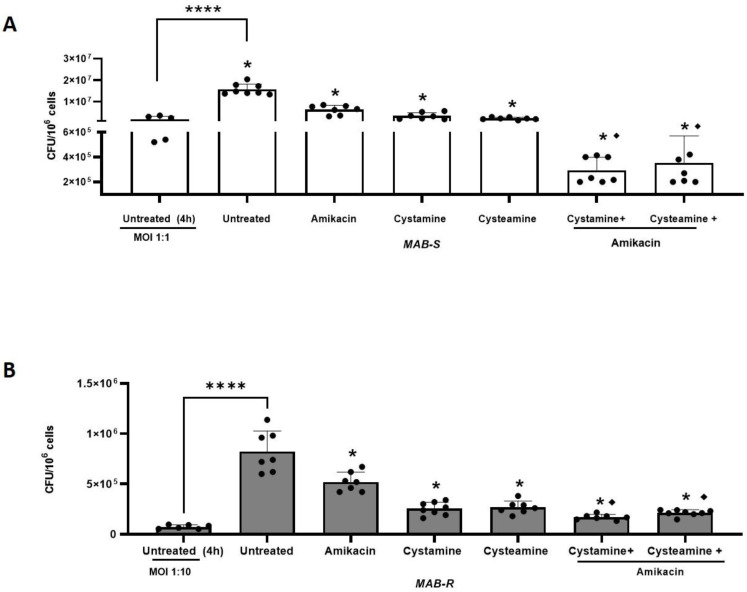
Enhanced effect of cysteamine/cystamine with amikacin. In (**A**,**B**), the THP-1 were infected with the R and S variants, respectively (the treatments and combinations with amikacin were added as described above). Two days after infection, cells were lysed to assess intracellular CFUs and the results are shown as Log CFU/10^6^ cells. Box plots chart represented the mean and SD of three independent experiments. Data were analyzed by one-way ANOVA followed by the Tukey comparison test. * represents statistically significant difference of all treatments compared with untreated condition (**** *p* < 0.001); ♦ represents statistically significant difference of cysteamine and cystamine in combination with amikacin compared with amikacin alone (**** *p* < 0.001 for both variants).

**Figure 3 ijms-24-01203-f003:**
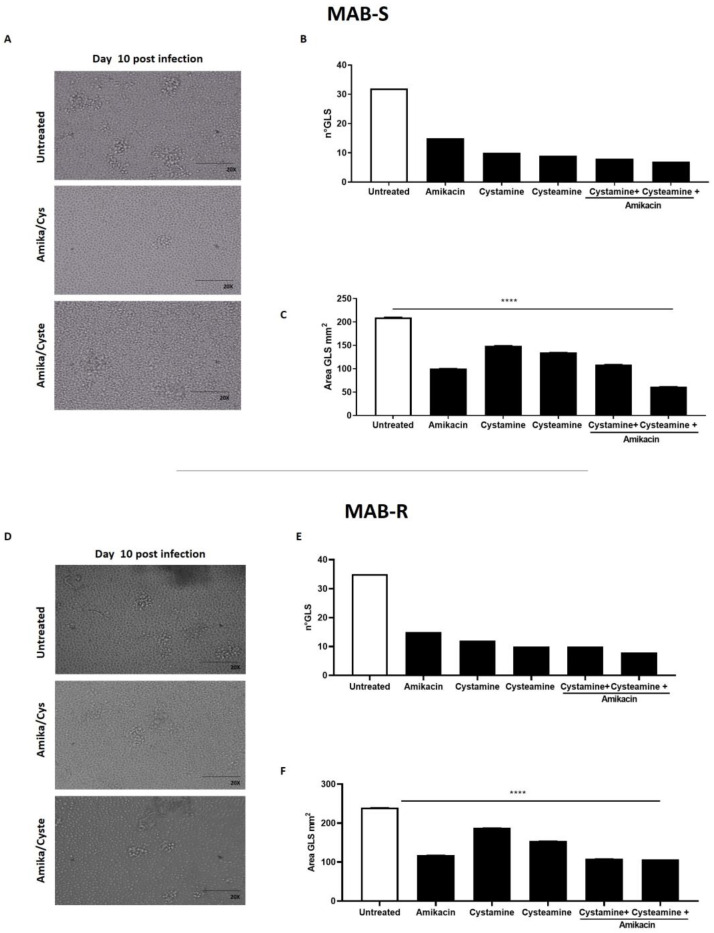
Evaluation of the anti-TG2 treatment in the GLS model of *M. abscessus* infection. PBMCs obtained from healthy donors were infected with the MAB-S and -R variants, respectively, and left to mature into GLSs for up to 10–12 days. GLSs were treated with different drugs starting at 3 days post-infection: cystamine (400 µM), cysteamine (800 µM), and amikacin (4 µg/mL). Representative light microscopy images of GLSs 10 days post-infection (magnification 20X) infected with MAB-S (**A**) and MAB-R (**D**). At this time point, assessing 12 fields/sample, we measured the number of the GLSs infected with MAB-S (**B**) and MAB-R (**E**) and the average surface of GLS in 12 fields (MAB-S (**C**) and MAB-R (**F**). The means ± standard deviations of scores representative of three experiments each are given the images; **** *p* < 0.0001 by one-way ANOVA with Dunnett’s multiple-comparisons test compared with untreated condition.

**Figure 4 ijms-24-01203-f004:**
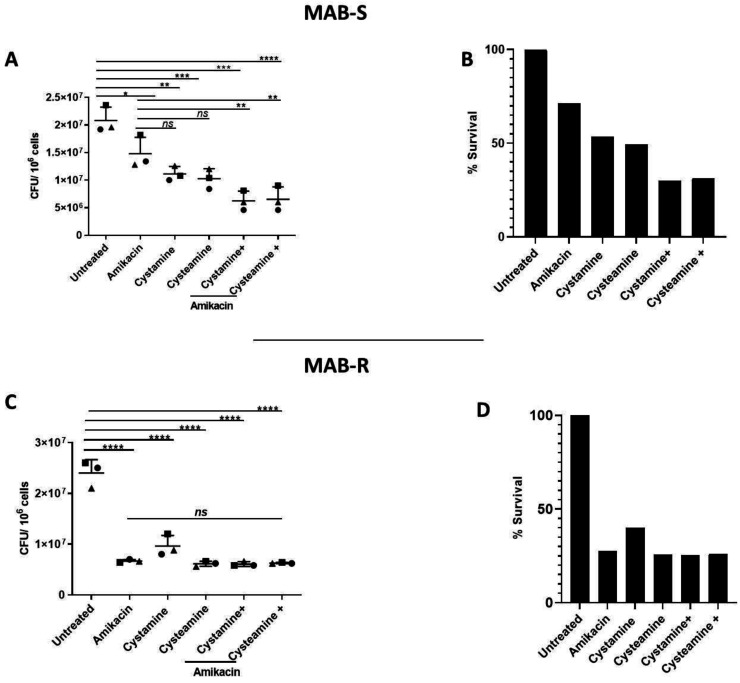
Enhanced effect of cysteamine and cystamine with aminoglycosides in GLS model. GLSs obtained from healthy donors were infected with MAB-S and -R variants at MOI 1:1 and 10:1, respectively, and then treated with different drugs starting at 3 days post-infection treatment of cysteamine and cystamine with amikacin (Sigma-Aldrich) (4 μg/mL). Total CFUs were determined at 12 days post-infection (**A**,**C**); * *p* < 0.05, ** *p* < 0.01, *** *p* < 0.005, **** *p* < 0.001 by one-way ANOVA and Tukey’s multiple comparisons test. Mycobacterial inhibition by the proposed treatments was expressed as a % of survival compared to the untreated group (**B**,**D**).

**Figure 5 ijms-24-01203-f005:**
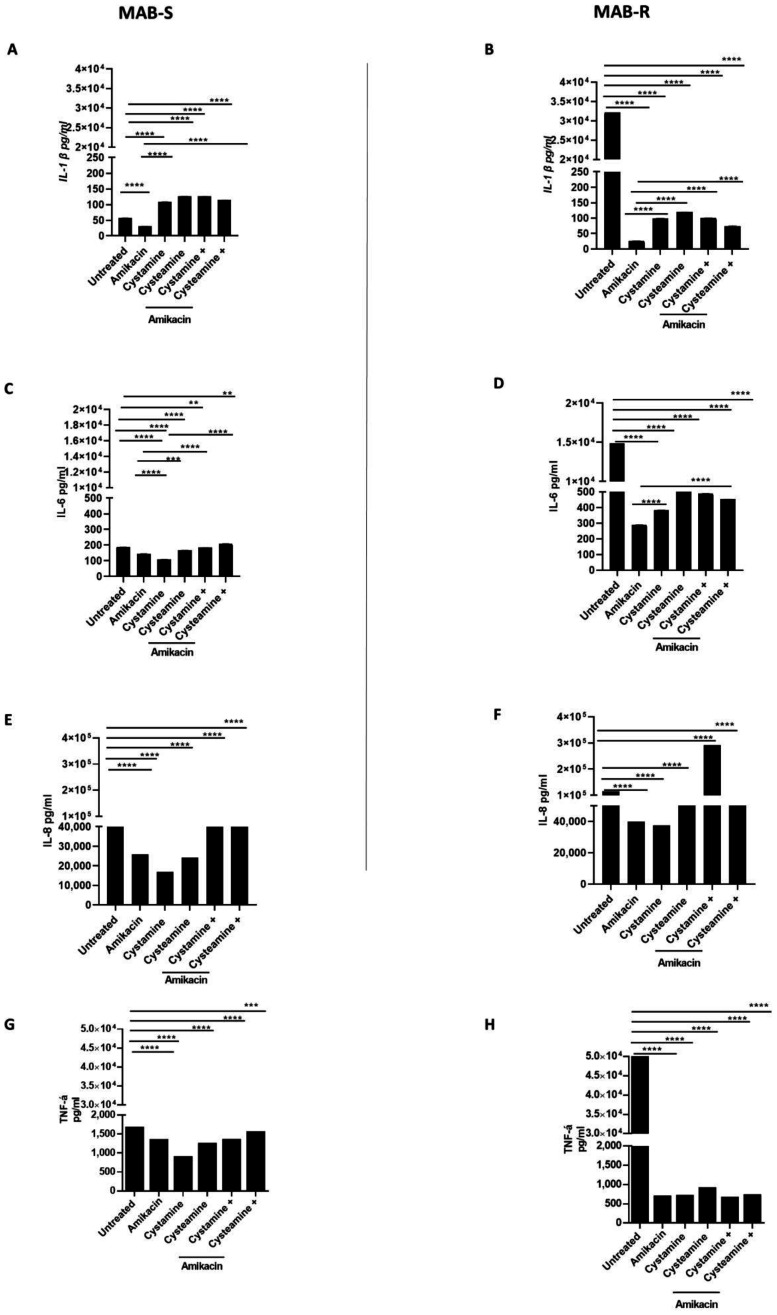
Comparison of pro-inflammatory cytokine production of MAB- R and S variants. Supernatants of PBMCs infected, as described above, with R and S variants were collected at 12 p.i and pro-inflammatory cytokine levels, IL-1β (**A**,**B**), IL-6 (**C**,**D**), IL-8 (**E**,**F**), TNF-α (**G**,**H**), were quantified by single-plex ELISA (Multi-cytokine test for ELLA, Bio-Techne). Supernatants harvested from uninfected PBMCs served as negative controls. Data are expressed as mean ± SD (pg/mL) from at least two independent experiments with at least three technical replicates. *p*- values were determined by one-way ANOVA and Tukey’s multiple comparisons test and ** *p* < 0.01, *** *p* < 0.005, **** *p* < 0.001.

**Table 1 ijms-24-01203-t001:** MIC determination * of the two *M. abscessus* variants, MAB R and MAB S, against a panel of antibiotics, in the presence or not of cystamine (400 µM) and cysteamine (800 µM). MIC are expressed as µg/mL.

	MAB R			MAB S		
Antibiotic	-	+Cystamine	+Cysteamine	-	+Cystamine	+Cysteamine
Clarithromycin	32	32	32	32	32	32
Rifabutin	8	8	8	8	8	8
Ethambutol	16	16	16	16	16	16
Isoniazid	8	8	8	8	8	8
Moxifloxacin	8	8	8	8	8	8
Rifampin	8	8	8	8	8	8
Trimethoprim/Sulfamet.	8/152	8/152	8/152	8/152	8/152	8/152
Amikacin	4	4	4	4	4	4
Linezolid	64	64	64	32	64	64
Cirpofloxacin	8	8	8	4	4	4
Streptomycin	64	64	64	64	64	64
Doxycyckine	16	16	16	16	16	16
Ethionamide	20	20	20	20	20	20

* Drug susceptibility testing (DST) for MAB R and MAB S was performed using the Sensititre SLOWMYCO system (Thermo Scientific™Ltd., Waltham, MA, USA).

## Data Availability

The original contributions presented in the study are included in the article/Appendix A. Further inquiries can be directed to the corresponding author.

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
