# Peer review of "Cysteamine/Cystamine Exert Anti-*Mycobacterium abscessus* Activity Alone or in Combination with Amikacin"

_ijms, 2023, doi:10.3390/ijms24021203_

Round 1

Reviewer 1 Report

Palucci and colleagues show efficacy of inhibiting transglutaminase-2 (TG2) enzyme as a host-directed therapy against M. abscessus infection. They test these FDA approved TG2 inhibitors-cysteamine and cystamine-in infected human macrophages and ex vivo granuloma model. The group has previously done a similar study using M. tuberculosis and observed similar efficacy.  

In its current form, the manuscript requires major revision. My comments are below:

1.       It has been shown previously that cysteamine has direct antimicrobial activity on polymicrobial populations, including M. abscessus (Devereux et al, 2015; Charrier et al, 2014). The authors do not observe this with the doses and time points tested in the current study. The authors must discuss this point in detail, and how their study is different. It would be important to test direct antimicrobial activity of cysteamine/cystamine for longer time points (10-12 days) in axenic culture in the current study.

2.       Line 255: the authors comment that the HDT restricts both Mab R and S variants similarly but at a superior level compared to amikacin. The effect of cysteamine/cystamine is not as pronounced on the S variant (~ half log CFU reduction) compared to R (at least a log reduction). Can the authors discuss why they think HDT efficacy is similar for both R and S variants? Can the authors show intracellular Mtb CFU at 4h time point in the same graph as 48h. This will help assess the growth pattern of M. abscessus R and S variants under the various conditions tested.

The authors have only tested one dose of amikacin, which is way lower compared to the HDTs. I am confused by their interpretation that the HDTs are superior to amikacin. In Fig 4C, Mab-R CFUs are similar across different treatments. Do the authors see maximal efficacy of amikacin at 4 ug/mL? What if they test higher doses of amikacin? Will the authors observe better CFU reductions? Perhaps a dose response curve would help here.

3.       Fig 1B is confusing. The authors mention using different MOIs (1 or 10) for infecting S and R variants. Can the authors explain how they see similar CFUs at 4 hour pi for these strains?

Also why is lower MOI used for S variant that grows slower intracellularly compared to R? Will HDT efficacy by different if higher MOI is used for Mab-S infections?

4.       Comparing figs 1C and 2A/B is confusing. The degree of inhibition by HDT doesn’t seem consistent between experiments. In fig1C, Mab-R CFU are reduced by ~1 log but in fig 2B reduction is barely half-log.  Were these conditions different?

5.       The authors attribute TG2 inhibition to reduced Mab infection.  However, in the PBMC model it is difficult to determine the degree of TG2 inhibition in different cell types present. With lack of clarity on the mechanism of HDT action it is difficult to ascertain how different cell types will respond for infection control. The authors must discuss this point as well.

Minor points:

1.       Typo line 268 change to Tukey

2.       Please add scale bars for all images in fig 3.

3.       Error bars missing in figs 4B, 4D, 5E, 5F.

Author Response

We are submitting the revised version of the manuscript entitled “Cysteamine/Cystamine exert anti-Mycobacterium abscessus activity alone or in combination with amikacin” , by Palucci I. et al, which has been revised according to the reviewer comments and suggestions. Below you will find the point-by-point rebuttal that specifically addresses all the reviewer’s concerns.

# Reviewer 1

Palucci and colleagues show efficacy of inhibiting transglutaminase-2 (TG2) enzyme as a host-directed therapy against M. abscessus infection. They test these FDA approved TG2 inhibitors-cysteamine and cystamine-in infected human macrophages and ex vivo granuloma model. The group has previously done a similar study using M. tuberculosis and observed similar efficacy. 

In its current form, the manuscript requires major revision. My comments are below:

  1. It has been shown previously that cysteamine has direct antimicrobial activity on polymicrobial populations, including M. abscessus (Devereux et al, 2015; Charrier et al, 2014). The authors do not observe this with the doses and time points tested in the current study. The authors must discuss this point in detail, and how their study is different. It would be important to test direct antimicrobial activity of cysteamine/cystamine for longer time points (10-12 days) in axenic culture in the current study.

We thank the reviewer for the important comment. In one of the papers mentioned (Charrier et al. 2014), cysteamine showed direct activity against relevant CF pathogens, with an MIC in the range of 250-500 ug/ml, that is around 3240-6480 uM, a concentration higher than that used in our experiments. In the second paper (Deveraux G et al 2015), using different experimental settings, cysteamine showed direct activity against polymicrobial loads in sputum at lower concentrations. However, the MIC for M. abscessus for cysteamine was in the range of 62,5-250 ug/ml, that is around 3240-810 uM, a concentration higher than that used in our experimental settings (800 uM). It is possible that differences in experimental settings and the M. abscessus strain used may explain the different results. We have tested the activity of cysteamine, and cystamine, alone or in combination with amikacin, in classical MIC experimental settings, in exponentially growing cultures (Fig. 1D-E) or in agar plates containing the drugs and seeded with M. abscessus. Since M. abscessus grows in less than 10-12 days, we think that it is not possible to adapt our experimental setting to longer time points. As suggested by the reviewer, we discuss these issues in the revised version of the manuscript.

  1. Line 255: the authors comment that the HDT restricts both Mab R and S variants similarly but at a superior level compared to amikacin. The effect of cysteamine/cystamine is not as pronounced on the S variant (~ half log CFU reduction) compared to R (at least a log reduction). Can the authors discuss why they think HDT efficacy is similar for both R and S variants? Can the authors show intracellular Mtb CFU at 4h time point in the same graph as 48h. This will help assess the growth pattern of M. abscessus R and S variants under the various conditions tested.

We thank for the comment. In Line 255, we comment that HDTs are more effective in quenching the inflammatory responses induced by the MAB R variant compared to the S variant. Conversely, the two HDTs exert a different antimycobacterial activity against the R and S variants depending on the experimental model used (THP-1 or PBMC), though we measured a significant activity against both variants. Cysteamine and cystamine enhance the intracellular antimicrobial responses of macrophages and host cells, thereby supporting the host cells in killing both forms following phagocytosis. We discussed this issue in the discussion (see also line 361). Moreover, we removed the term “superior”.

According with the reviewer's comments, we changed the figures 2 A and B, including the results of the CFUs determined at 4h p.i.

The authors have only tested one dose of amikacin, which is way lower compared to the HDTs. I am confused by their interpretation that the HDTs are superior to amikacin. In Fig 4C, Mab-R CFUs are similar across different treatments. Do the authors see maximal efficacy of amikacin at 4 ug/mL? What if they test higher doses of amikacin? Will the authors observe better CFU reductions? Perhaps a dose response curve would help here.

We thank for the comment that offers us the opportunity to clarify one important issue. To investigate the potential usefulness of HDTs against M. abscessus infections, we reasoned that it would have been useful to compare the activity of the HDTs with that of a commonly used antibiotic administered at the MIC. In this way, we mimicked the activity of the HDTs in supporting “minimally” active antibiotics, which is a quite common condition during the treatment of M. abscessus and other NTM infections. We clarified this issue in the revised version of the manuscript. Hence, we administered amikacin at the concentration of 4 ug/ml, that corresponds to the MIC for the M. abscessus used (Table 1), as determined with the SLOWMYCO system Sensititre 96. Again, we removed the term “superior”.

  1. Fig 1B is confusing. The authors mention using different MOIs (1 or 10) for infecting S and R variants. Can the authors explain how they see similar CFUs at 4 hour pi for these strains?

Also why is lower MOI used for S variant that grows slower intracellularly compared to R? Will HDT efficacy by different if higher MOI is used for Mab-S infections?

We thank for the comment. As also described in previous reports (Byrd TF, Lyons CR. 1999; Nessar R, Reyrat J-M, Davidson LB, Byrd TF. 2011; Anne-Laure Roux,et al 2016), the two MABs variants show different pathogenetic and virulence properties. In fact, the MAB R variant is more virulent compared to the S variant, as previously observed by other authors (Brambilla C Llorens-Fons 2016, Etienne et al., 2002; Villeneuve et al., 2003; Kocíncová et al., 2009). Hence, we “balanced” the MOI for the two variants to allow a productive infection while avoiding excessive cell damage. Therefore, we used a higher MOI when infecting with the MAB S variant. These MOI levels yielded between 50 and 60% of macrophages infected at 4 h.p.i.

  1. Comparing figs 1C and 2A/B is confusing. The degree of inhibition by HDT doesn’t seem consistent between experiments. In fig1C, Mab-R CFU are reduced by ~1 log but in fig 2B reduction is barely half-log. Were these conditions different?

Thanks for the comment. This is actually a very relevant aspect that we addressed in the revised version of the manuscript. The results shown in Figure 1C and in Figure 2A/B are not consistent because in Fug. 1C we infected with the same MOI for the R and S variant, while in Fig. 2A/B we used the “balanced” MOI (see previous comment).  Indeed, we think that these two apparently contradictory results may be confusing. For these reasons, we removed Fig. 1C and presented the results of the MAB S and MAB R variants only in Fig. 2A/B, with infections carried out at two different MOIs. Moreover, we included the results of the CFUs determined at 4h p.i. for the two variants, as suggested by the reviewer. We thank the reviewer for the suggestion.

  1. The authors attribute TG2 inhibition to reduced Mab infection. However, in the PBMC model it is difficult to determine the degree of TG2 inhibition in different cell types present. With lack of clarity on the mechanism of HDT action it is difficult to ascertain how different cell types will respond for infection control. The authors must discuss this point as well.

We thank for the comment. We addressed in the discussion the potential activity of the two HDTs cysteamine and cystamine during infection with Mab, where they can enhance the intracellular antimicrobial activity of host cells or modulate inflammatory responses that in turn have an impact on the ability of macrophages and other cells found in PBMCs to contain mycobacterial replication. A specific characterization of the activity of the two HDTs on different cell types goes beyond the scope of this study and we hope that the results provided in this manuscript can stimulate further studies on this topic. We have specifically addressed this point in the revised version of the manuscript. (discussion section, line …)

Minor points:

  1. Typo line 268 change to Tukey

We thank for the comment. We corrected  mistake.

  1. Please add scale bars for all images in fig 3.

We thank for the comment. We revised the figures as suggested.

  1. Error bars missing in figs 4B, 4D, 5E, 5F.

We thank for the comment. In the case of figure 4B/D, the bar represented a % of survival; instead, are present but 2 error bars are clipped at the axis limit

(Please see the attachment).

Reviewer 2 Report

Throw somewhat more light on the pathogenicity features of the three NTM species, Mycobacterium avium complex, Mycobacterium intracellulare complex and M. abscessus to reach broader understanding of the clinicopathological significance of the presented work.

Author Response

We thank the reviewer for the supportive comments.

Round 2

Reviewer 1 Report

The authors have addressed my concerns. I have no further comments.